# Unsupervised Feature Fusion Model for Marine Raft Aquaculture Sematic Segmentation Based on SAR Images

1st Mengmeng Li
*School of Information Science and Engineering*
*Dalian Polytechnic University*
Dalian, China
220520854000601@xy.dlpu.edu.cn

2nd Xinzhe Wang
*School of Information Science and Engineering*
*Dalian Polytechnic University*
Dalian, China
wxzagm@dlpu.edu.cn

3rd Jianchao Fan *
*School of Control Science and Engineering*
*Dalian University of Technology*
Dalian, China
fjchao@dlut.edu.cn

*Abstract*—**Marine aquaculture semantic segmentation provides a scientific basis for marine regulation and plays an important role in marine ecological protection and management. Currently, most high-performance marine aquaculture segmentation networks are trained by supervised learning. This approach requires collecting a large number of accurate manually labelled samples for training, but the labelled samples are difficult to obtain. To solve this problem, this paper proposes an unsupervised feature fusion model (UFFM) for marine raft aquaculture semantic segmentation. Firstly, a pseudo-label generator is designed to label the training samples, and a coarse mask is generated using saliency feature clustering. The training samples with pseudo-labels are inputted into a multilevel feature fusion module to extract further and continuously improve the graphical shapes and categories of the objects under the guidance of cross-entropy loss. The pseudo-labels are optimised under continuous iteration to improve the model segmentation performance. Comparison experiments on the GF-3 dataset demonstrate the effectiveness of UFFM.**

*Index Terms*—**unsupervised learning, pseudo-label, SAR images, semantic segmentation**

## I. INTRODUCTION

China has witnessed rapid growth in the scale and benefits of marine aquaculture development in recent years [1]. However, while the marine aquaculture industry has made significant progress, it is also faced with problems such as pollution around aquaculture waters, irrational layout of aquaculture, and excessive density of offshore aquaculture [2].

This work was supported in part by the National Natural Science Foundation of China under Grant 42076184, Grant 41876109, and Grant 41706195; in part by the National Key Research and Development Program of China under Grant 2021YFC2801000; in part by the National High Resolution Special Research under Grant 41-Y30F07-9001-20/22; in part by the Fundamental Research Funds for the Central Universities under Grant DUT23RC(3)050; and in part by the Dalian High Level Talent Innovation Support Plan under Grant2021RD04. (Corresponding author: Jianchao Fan.)

Synthetic aperture radar (SAR) has the advantage of being all-weather and does not need to consider factors such as cloudy weather. It has become an essential tool for monitoring marine aquaculture. The backscattering features of the mariculture raft target in SAR images are much larger than the backscattering features of the seawater surface, which makes the aquaculture rafts and seawater background present a high contrast [3]. Researchers have adopted deep learning techniques to design various mariculture semantic segmentation methods to efficiently and accurately extract the mariculture information [4].

However, existing neural network models usually rely on a large amount of manually labeled data for training to obtain high-accuracy results. This approach faces two main problems: 1) the cost of obtaining high-quality manually labeled data is extremely high in complex scenarios and when dealing with massive remote sensing data, resulting in a large amount of remote sensing data that cannot be fully utilized. 2) the reliance on manual labelling as the only learning signal leads to limited feature learning. Several studies have proposed unsupervised methods for extracting information on marine aquaculture to address these challenges. Fan et al. [3] proposed using the multi-source characteristics of floating rafts and combining the neurodynamic optimization with the collective multi-core fuzzy C-means algorithm to classify unsupervised aquaculture. Wang et al. [5] designed an incremental dual unsupervised deep learning model based on the idea of alternating iterative optimization of pseudo-labels and segmentation results to maintain and strengthen the edge semantic information of pseudo-labels and effectively reduce the influence of coherent spot noise in SAR images. Subsequently, Zhou et al. [6] constructed an unsupervised semantic segmentation network for mariculture based on mutual information theory and superpixel algorithm, which improves the continuity and spatial consistency of mariculture target extraction through global

feature learning, pseudo-label generation, and optimization with mutual information loss. However, the above unsupervised deep learning models mainly rely on single-area data training, which is difficult to generalize to intelligent image interpretation in wide-area and complex scenes.

With the emergence of transformer [7], a self-supervised representation learning model using unlabeled remote sensing big data to address regional feature differences. Self-supervised transformer network can learn its spatial features from a large amount of remote sensing data by constructing a pretexting task and pre-training the vision transformer model, which applies to a variety of downstream tasks by fine-tuning, e.g., change detection [8], classification [9], target detection [10], and semantic segmentation tasks [11]. Fan et al. [12] established a self-supervised feature fusion transformer model to obtain the essential features of mariculture through a large number of unlabeled samples, introduced contrast loss and mask loss, and paid attention to the global and local features of aquaculture at the same time, which mitigated the problems of mutual interference among multiple targets and imbalance of data between classes, and realized the accurate segmentation of mariculture. However, the self-supervised transformer model can rely on a large number of unlabeled floating raft aquaculture data for information extraction on a single sea area but still needs high-quality labeled data fine-tuning in the downstream segmentation network.

To solve the above problems, this paper applies the saliency information obtained from self-supervised representation learning to the downstream segmentation network. It combines it with a multi-stage feature fusion module to further enhance the semantic segmentation performance of the network. Specifically, a pseudo-label generator is first designed to generate saliency pseudo-labels. Then, the semantic segmentation results output by the multilevel feature fusion module is cross-entropy loss with the pseudo-labels, which are constrained and directionally passed parameters to the network. The pseudo-labels are optimised through continuous iteration to improve network segmentation performance further.

## II. RELATED WORK

### A. Self-supervised feature learning

Self-supervised learning mainly utilizes auxiliary tasks to mine supervised information from large-scale unmanually labeled data. It trains the network with this constructed supervised information to learn valuable representations for downstream tasks. Common auxiliary tasks include comparative learning, generative learning, and comparative generative methods that design learning paradigms based on data distribution characteristics to obtain better feature representations. However, these methods are mainly focused on image classification tasks and thus are typically designed to generate separate global vectors from images as input. This problem leads to poor results downstream for densely predicted segmentation tasks, requiring high-quality truth-labeled fine-tuned models. However, the emergence of self-supervised transformer has

made it possible to extract dense feature vectors without requiring specialized dense contrast learning methods, which can reveal hidden semantic relationships in images. In this paper, inspired by DINO [13], the upstream trained image saliency features generate pseudo-labels for the training data to fine-tune the downstream segmentation network to construct a fully unsupervised semantic segmentation model.

### B. Unsupervised semantic segmentation

Unsupervised semantic segmentation aims at class prediction for each pixel point in an image without artificial labels. Ji et al. [14] proposed invariant information clustering (IIC), which ensures cross-view consistency by maximising the mutual information between neighbouring pixels of different views. Li et al. [15] constructed PiCIE to learn the invariance and isotropy of photometric and geometric variations by using geometric consistency as an inductive bias. This approach is that it only works on dataset MS COCO, which does not distinguish between foreground and background classes. MaskContrast [16] first generates object masks using DINO pre-trained ViT and then uses pixel-level embeddings obtained from contrast loss. However, the method can only be applied to saliency datasets. For the multi-stage paradigm, researchers tried to utilise class activation maps (CAM) [17] to obtain initial pixel-level pseudo-labels, which were then refined using a teacher-student network. However, this would result in losing features during training, decreasing segmentation accuracy. In this paper, to solve the above problems, Grad-CAM [18] is introduced in multi-stage to generate pseudo-labels and improve the segmentation performance by multi-scale feature fusion.

## III. METHOD

### A. Overall framework

In the upstream task, a large amount of unlabeled marine aquaculture data is trained from zero to obtain the pre-trained ViT weights $\theta_t$ and initialize the downstream feature extraction network. The overall architecture designed for the downstream segmentation task is shown in Fig. 1. The processed unlabeled marine aquaculture images are used as inputs to the network to obtain the segmentation results of the aquaculture. The designed network have two branches, one for generating pseudo labels using saliency features and the other is a segmentation branch for multi-layer feature fusion. First, in the upstream task, large-scale unlabeled data is used to pre-train the ViT [13] in order to obtain the initialization parameters $\theta_t$ of the downstream feature extraction network, which can accelerate the convergence of the downstream segmentation network by using the pre-training weights and is crucial for the extraction of the model to salient features. The designed network is shown in Fig. 1. First, an input unlabeled marine aquaculture image, which has been stretched in a linear phase and rotated randomly, is used to augment the original image with data. The input image will go through two branches: one is the saliency pseudo-label generation branch, which will be presented in III-B, and the other is the multi-layer transformer feature

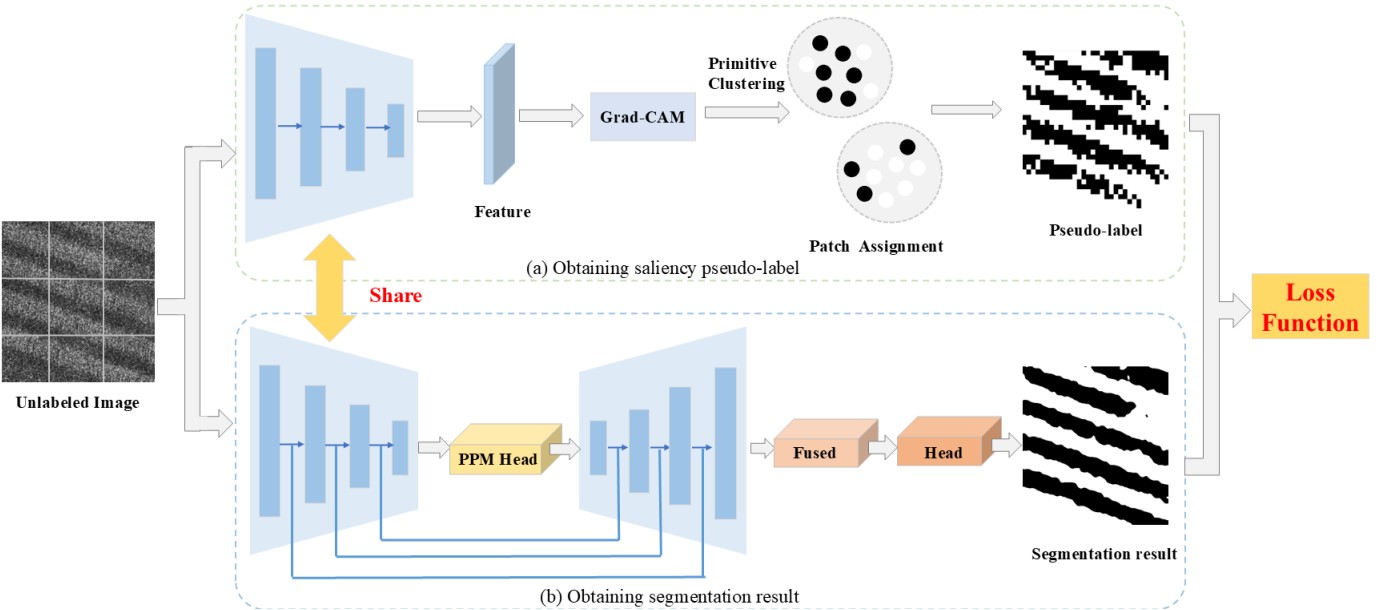

Fig. 1. Overview model of UFFM. (a) Obtaining saliency pseudo-label: Input the multi-head self-attention mechanism of the last layer feature map in the transformer block into Grad-CAM to obtain saliency patch features and generate saliency pseudo-label. (b) Obtaining segmentation results: The semantic information is enhanced using a multilayer transformer with PPM, and the semantic segmentation results with pseudo-labels are output by backpropagation after the loss computation. After continuous iterative updates, the network segmentation performance is improved.

fusion branch, which will be presented in III-C. In network training, the supervisory loss $\mathcal{L}_s$ is the pixel-by-pixel cross-entropy loss between the pseudo-labeled pixel level and the prediction:

$$\mathcal{L}_s = \frac{1}{N} \sum_{i=0}^{N-1} CrossEntropy(\tilde{y}_i, y_i) \qquad (1)$$

where $N$ denotes the number of pixels in the image $x \in \mathbb{R}^{H \times W \times 3}$ and $y_i \in \mathbb{R}^C$ is the network's prediction probability for pixel $i$, where $C$ is the number of predicted classes and $\tilde{y}_i \in \mathbb{R}^C$ is the labelling class of pixel $i$ in the pseudo-label.

During the network training, the loss will be gradient back to the feature extraction network, and in particular, the weights of the two branches will be shared and updated simultaneously. Through continuous iteration of the network, the pseudo-label is updated, thus improving the segmentation performance of the network.

### B. Saliency pseudo-label generation

In unsupervised tasks, the design of pseudo-labels is crucial. A simple approach is to use confidence thresholds followed by direct results output as pseudo-labels. However, this approach is unsatisfactory in processing complex data and produces poor results. To solve this problem, a variant of activation graph-like Grad-CAM is used in this paper to generate significance discriminative pseudo-labels by stepwise subdivision from the target localisation method. Given an image $x$, generate a sequence of patch embeddings $x_{patch} \in \mathbb{R}^{P \times D}$, where $P$ is the number of patches, and $D$ is the output dimension. Then, $x_{CLS} \in \mathbb{R}^{1 \times D}$ and position embedding P are also added to the concatenated inputs. Therefore, the input sequence $z_0$ of ViT is described as:

$$z_0 = [x_{patch}, x_{CLS}] + \mathrm{P} \qquad (2)$$

After that, the last layer of features is obtained through multiple layers of transformer encoders. The saliency feature map is computed using Grad-CAM. The first k salient patches with the largest absolute value of the gradient of the embedded image patch features are selected as the salient patches, and finally, a binary operation is performed to mark the first $k$ salient patches as 0 and the rest as 255. The generated saliency pseudo-label $\tilde{y}$ is written as:

$$g_k = Sum \left| \frac{\partial L\left(f(x), y\right)}{\partial x_{patch}^k} \right| \qquad (3)$$

$$\tilde{y} = \begin{cases} 0, & \text{if } g_k \text{ in } topk\mathrm{G} \\ 255, & \text{otherwise} \end{cases} \qquad (4)$$

where $\mathrm{G} \in \mathbb{R} = \{g_1, g_2, \ldots g_K\}$ is the salience map of patches $x_{patch} = \left\{x_{patch}^1, \ldots x_{patch}^K\right\}$ topk is the set of selected salient patches.

### C. Multi-stage feature fusion

The segmentation decoder consists of a pyramid pooling module (PPM) and a multi-scale feature pyramid to enable the network to capture contextual semantic information better. Firstly, three feature maps $\{V_2, V_3, V_4\}$ are generated at the transformer encoder. The output feature vectors are the same size since the model chosen is the base ViT model, and the last transversal $L_5$ is generated from the last feature map

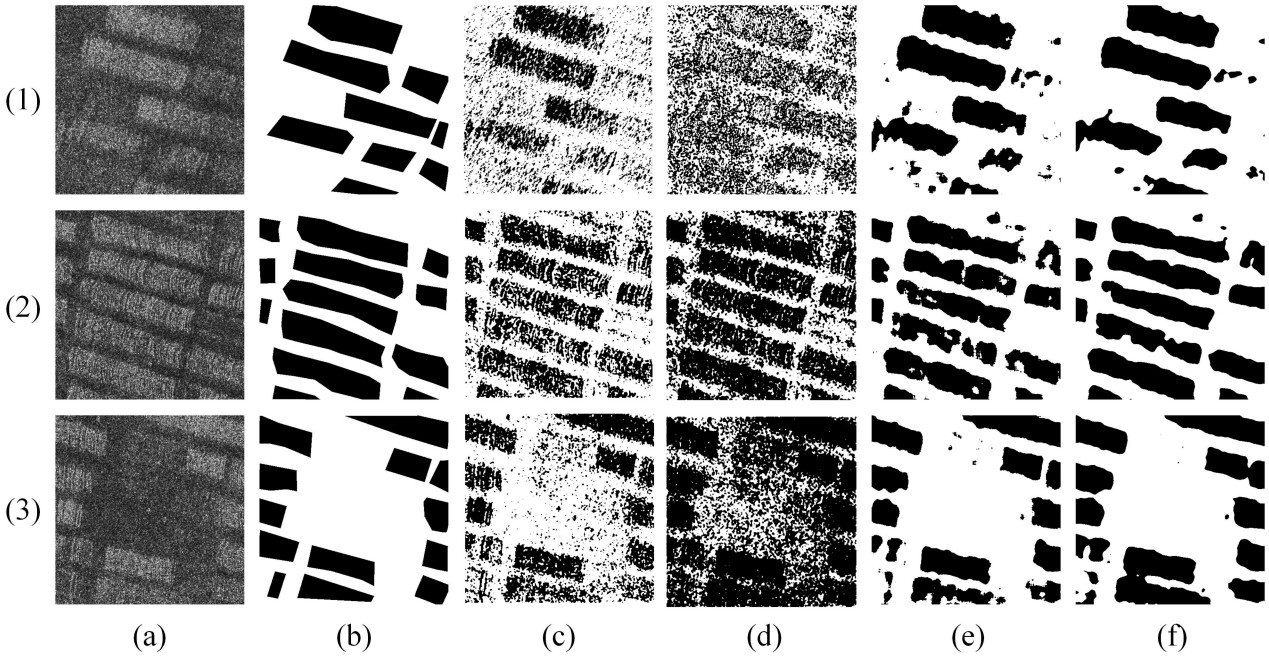

Fig. 2. Visual comparison of raft marine aquqculture segmentation on the GF-3 dataset. (a) original images. (b) ground-truth labels. (c) IIC. (d) PiCIE. (e) IDUDL. (f) UFFM.

$V_5$ through the PPM module. The FPN sub-network then paths down from the top to the branch to obtain $F_i = L_i + UP_2(F_{i+1})$, $i = \{2, 3, 4\}$, where the operation Up denotes bilinear upsampling. The FPN then uses the convolutional block $h_i$ to obtain the output $P_i$ respectively. The final feature fusion of the FPN output requires bilinear upsampling of each po to ensure that they have the same spatial size and is finally connected by the channel dimension and fused by the convolutional unit block $h$

$$Z = h\left([P_2; UP_2(P_3); UP_4(P_4); UP_8(P_5)]\right) \quad (5)$$

The fused feature Z is then subjected to 1×1 convolution and 4× bilinear upsampling to obtain the final prediction $y$.

## IV. EXPERIMENTAL RESULTS

### A. Experiment Setup and Datasets

All experiments are conducted in PyTorch 1.8.1, using an Intel Xeon Platinum 8255C with a clock speed of 2.5 GHz and an Nvidia GeForce RTX 3090. The data enhancement strategy was consistent with DINO [13]. A vit - s /16 model [7] trained with self jitter loss was used to extract features from the patches. The learning rate was set to 0.05. In addition, a stochastic gradient descent (SGD) optimiser with a momentum of 0.9 was used. The encoder part uses ViT as the main network. The decoder part uses the UPerHead architecture to receive features from all levels of the encoder and generate the final prediction through pooling and upsampling operations. Meanwhile, the auxiliary head uses FCNHead architecture to receive features from specific encoder layers.

The study area is located in the sea water aquaculture area of Changhai County, China. The remote sensing images were preprocessed with radiometric calibration and geographic correction, and the remote sensing images with horizontal-horizontal(HH) polarisation mode are selected as the experimental data. The images are subsequently cropped to $512\times512$ pixels. The self-supervised pre-training of the GF-3 dataset is more than 13,000, the downstream train datasets is 369, and the test datasets is 160.

### B. Evaluation Metrics

In SAR images, there are a large number of coherent spot noise effects on raft aquaculture targets, resulting in a large number of isolated noise points in the image, which affects the accurate extraction of raft aquaculture targets. Therefore, in this paper, multiple evaluation metrics are used to evaluate the segmentation results. The metrics refer to IDUDL, which contains mIoU ($mIoU$), Kappa coefficient ($K$), Overall Accuracy ($OA$), Precision ($P$), Recall ($R$) and F1 score ($F_1$).

Where $mIoU$ evaluates the average degree of overlap between the predicted pixel categories and the true value pixel categories, which enables a better evaluation of the semantic continuity and consistency of the model predictions. $K$ considered the effect of chance coincidences when evaluating the degree of consistency. $OA$ evaluates the proportion of correctly predicted pixel classes in the overall correctly predicted pixel classes, reflecting the global accuracy. $P$ denotes the proportion of float samples predicted by the model. $R$ represents the ability of the model to find all positive samples. $F_1$ synthesis balances $P$ and $R$.

TABLE I
QUANTITATIVE COMPARISON OF PROPOSED WITH OTHER UNSUPERVISED
DEEP LEARNING METHODS ON THE SAME DATASET. THE BEST RESULTS
ARE HIGHLIGHTED AS BOLD.

| Methods | $mIoU$ | $Kappa$ | $OA(\%)$ | $P(\%)$ | $R(\%)$ | $F1$ |
|---|---|---|---|---|---|---|
| IIC [14] | 0.4613 | 0.2375 | 70.95 | 72.76 | 89.60 | 0.8063 |
| PiCIE [15] | 0.4905 | 0.3504 | 68.73 | 80.98 | 70.60 | 0.7198 |
| IDUDL [5] | 0.6102 | 0.5364 | 78.46 | 83.07 | **91.34** | 0.8130 |
| UFFM | **0.6371** | **0.5890** | **79.44** | **91.74** | 75.30 | **0.8371** |

### C. Comparison Results for Semantic Segmentation

Two classical unsupervised deep learning IIC [14] methods with PiCIE [15] method and an unsupervised deep learning model IDUDL [5] specifically designed for marine aquaculture are selected. The semantic segmentation results of different methods are shown in Table I. The results showed that the proposed method improved the $mIoU$ by 0.0269 compared to IDUDL, while $P$ increased by 8.67%.

The visualisation results are shown in Fig. 2. In Fig. 2, the proposed method performs better in continuity and can reduce the interference of coherent spot noise. The effect of coherent spot noise in SAR images leads to many bright noises, affecting the segmentation results. The method of utilising mutual information in IIC can enhance the degree of correlation between similar samples. However, the noisy pixels are still strongly correlated with the target pixels, which leads to the impossibility of removing a large number of noisy pixels in the segmentation results.PiCIE utilises the method of geometric invariance and photometric invariance to maintain semantic consistency, but a large number of misclassifications occur. IDUDL can extract semantic features, overcome many noisy pixels, and perform the floating boundary better. However, the lack of global information leads to many missed judgments. Sample (2) shows that the proposed method can reduce the underdetermination in rafting compared to IDUDL.

## V. CONCLUSION

This paper proposes a new unsupervised feature fusion model, UFFM, for marine raft aquaculture semantic segmentation based on SAR images. The saliency obtained from representational learning generates saliency pseudo-labels in the pseudo-label generator. During network training, multi-stage feature fusion is designed to enhance the semantic information and the extraction of raft aquaculture target boundaries and semantic continuity. The experimental results show that UFFM can effectively reduce the problem of omission and misjudgment of raft aquaculture targets.

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
