# OpenReview forum: "Unsupervised Feature Fusion Model for Marine Raft Aquaculture Sematic Segmentation Based on SAR Images"
_IEEE.org/ICIST/2024/Conference — IEEE ICIST 2024 Conference Submission_

### Official Review · Reviewer_MPdp · 2024-08-21
**A work with a certain degree of innovation**

**Rating:** 7
**Confidence:** 4

**Review:**

The paper titled "Unsupervised Feature Fusion Model for Marine Raft Aquaculture Sematic Segmentation Based on SAR Images" proposes an unsupervised feature fusion model (UFFM) for marine raft aquaculture semantic segmentation. Firstly, a pseudo-label generator is proposed to label the training samples. And The pseudo-labels are optimised under continuous iteration to improve the model segmentation performance. Finally, the comparison experiments are proposed to illustrate the effectiveness of the proposed method. My specific feedback is as follows:
1) What role does semantic segmentation of marine aquaculture play in actual ocean regulation? Relevant background introduction can enhance the practical significance of the article.
2) Some formatting issues need to be addressed.

---

### Official Review · Reviewer_dsaW · 2024-08-21
**Accept**

**Rating:** 7
**Confidence:** 3

**Review:**

This paper proposed an unsupervised feature fusion model (UFFM) for marine raft aquaculture semantic segmentation. The theory is correct and can be accepted after responding the following comments.
(1)	In the introduction, it is not enough to state the current work. It should be expended and reconstructed.
(2)	There are many typos and grammar errors. The authors should have a native English speaker or software packages to perform the editing check.
(3)	At the end of the manuscript, the authors should further explain the future research direction in view of the proposed control strategy and obtained results, and illustrate its practicability in application in the conclusion.

---

### Official Review · Reviewer_vwGs · 2024-08-23
**Marine aquaculture sematic segmentation provides a scientific basis for marine regulation and plays an important role in marine ecological protection and management. Currently, most high-performance marine aquaculture segmentation networks are trained by supervised learning. This approach requires collecting a large number of accurate manually labelled samples for training, but the labelled samples are difficult to obtain. To solve this problem, this paper proposes an unsupervised feature fusion model (UFFM) for marine raft aquaculture semantic segmentation. Firstly, a pseudo-label generator is designed to label the training samples, and a coarse mask is generated using saliency feature clustering. The training samples with pseudolabels are inputted into a multilevel feature fusion module to further extract and continuously improve the graphical shapes and categories of the objects under the guidance of cross-entropy loss. The pseudo-labels are further optimised under continuous iteration to improve the model segmentation performance. Comparison experiments on the GF-3 dataset demonstrate the effectiveness of UFFM. Comments for this submission are given as follows.**

**Rating:** 7
**Confidence:** 3

**Review:**

(1)The grammar of this article is very good, the article is well worded, but individual areas need further work and the writer should double check the grammar of this article.
(2)Table I clearly states the results of the quantitative comparison of this paper with other unsupervised deep learning methods, please explain this further.
(3)The conclusion of this article is very well written, the article is well worded, but it does not describe the next work, the author should give a brief summary of the next work at the end.

---

### Decision · Program_Chairs · 2024-09-06

Accept (Oral)